# Development of an intraductal papillary mucinous neoplasm malignancy prediction scoring system

**Masanori Kobayashi**[1]*, **Hiromune Katsuda**[1], **Aya Maekawa**[2], **Keiichi Akahoshi**[2], **Ryosuke Watanabe**[3], **Yuko Kinowaki**[4], **Hisaaki Nishimura**[5], **Takeo Fujiwara**[5], **Minoru Tanabe**[2], **Ryuichi Okamoto**[1]

1 Department of Gastroenterology and Hepatology, Tokyo Medical and Dental University (TMDU), Tokyo, Japan, 2 Department of Hepato-Biliary-Pancreatic Surgery, Tokyo Medical and Dental University (TMDU), Tokyo, Japan, 3 Department of Diagnostic Radiology and Nuclear Medicine, Tokyo Medical and Dental University (TMDU), Tokyo, Japan, 4 Department of Comprehensive Pathology, Graduate School of Medical and Dental Sciences, Tokyo Medical and Dental University (TMDU), Tokyo, Japan, 5 Department of Global Health Promotion, Tokyo Medical and Dental University (TMDU), Tokyo, Japan

* mkobayashi.gast@tmd.ac.jp

**Data Availability Statement:** All relevant data are within the manuscript and its Supporting Information files.

## Abstract

Despite the presence of various guidelines, diagnosing malignant intraductal papillary mucinous neoplasm (IPMN) continues to pose challenges. Furthermore, although endoscopic ultrasonography (EUS) offers high-resolution images, it has not yet recognized as the primary tool for malignancy diagnosis. The study objective was to develop a simplified and user-friendly scoring system to improve the diagnostic accuracy of malignant IPMNs. Additionally, the utility of EUS and its effect on diagnostic accuracy were assessed. We retrospectively collected the clinical data on 160 cases of resected IPMN at Tokyo Medical and Dental University Hospital from January 2008 to December 2022. We examined clinical features, computed tomography (CT) and magnetic resonance imaging (MRI) findings, and EUS results if available. We then calculated the odds ratio of malignancy for these factors and developed an IPMN malignancy prediction (IMAP) scoring system. There were 89 (55.6%) cases of benign IPMNs and 71 (44.4%) of malignant IPMNs. Eight clinical and imaging findings, including age, diabetes mellitus status, jaundice, CA19-9 level, enhancing mural nodules ≥5mm, thickened wall, and main duct dilatation, were significantly associated with malignancy. The IMAP score was calculated by assigning 0 to 2 points to these factors based on the odds ratio. The area under the receiver operating characteristic curve for the IMAP score was 0.78 [95% confidence interval (CI): 0.71–0.85] based on CT/MRI alone and improved to 0.81 (95% CI: 0.74–0.87) when EUS was added. When the total exceeds 5 points, the positive predictive value becomes 100% (95% CI: 95.9–100). In conclusion, the IMAP scoring system has demonstrated promise as a clinically useful tool, offering both simplicity and sufficient accuracy. It holds potential as an important decision criterion for determining the treatment approach for IPMN. Additionally, EUS contributes to enhancing the diagnostic accuracy of the IMAP scoring system, thereby enabling more precise decision-making.

**Funding:** The author(s) received no specific funding for this work.

**Competing interests:** The authors have declared that no competing interests exist.

## Introduction

Intraductal papillary mucinous neoplasm (IPMN) is a premalignant pancreatic lesion that has a risk of carcinogenesis, with an estimated frequency of approximately 3%-8% [1–3]. There are several guidelines for the management of pancreatic cystic lesions, particularly IPMN. The International Consensus Guideline (ICG), European evidence-based guideline (European EBG), and American Gastroenterological Association institute guideline (AGA IG) are most commonly cited [4–6]. Although these guidelines commonly mention cyst size, main duct size, and enhancing mural nodules or masses as imaging factors that may indicate malignancy, it remains challenging to predict malignancy on the basis of a single parameter. Both computed tomography (CT) and magnetic resonance imaging (MRI) studies are recommended for better characterization in IPMN surveillance. However, due to considerations regarding radiation exposure, MRI is usually the preferred imaging modality. On the other hand, endoscopic ultrasonography (EUS) provides more detailed imaging of pancreatic cysts than can be obtained by CT or MRI, without exposing patients to radiation. In the revised International Consensus guideline for this year, there is mention of the usefulness of EUS, but it remains that EUS is not the main diagnostic tool [6]. Presently, there is no definitive evidence regarding the specific imaging findings from particular studies that are deemed essential for diagnosing malignant IPMN [7–9]. Therefore, even when following these guidelines, the diagnosis of malignant IPMN is not straightforward.

Consequently, scoring systems have been developed to improve the diagnostic accuracy of malignant IPMN by predicting the probability of malignancy [10–19]. Nevertheless, no model has been as widely adopted as the Child-Pugh classification in liver cirrhosis for the following reasons. To improve the diagnostic accuracy, models often employ complex mathematical formulas and establish cutoff values that deviate from the guidelines, particularly in terms of main duct size and nodule size [13, 18], so it becomes necessary to memorize the values for clinical use, which can be inconvenient. Other models can achieve high diagnostic accuracy by directly inputting values into a computer [15–17], eliminating the need to remember cutoff values or assigned scores. Nevertheless, the necessity to individually input values into a computer also might be inconvenient in clinical settings.

The study objective was to develop a diagnostic scoring system with a high level of accuracy for malignant IPMN. High accuracy was achieved by collecting detailed patient histories, which are only obtainable in a single-center study, and performing image evaluations based on standardized criteria within the institution. The development of the scoring system focused on ensuring clinical understanding and ease of memorization. Additionally, we used this scoring system to investigate the utility of EUS in the diagnosis of malignant IPMN.

## Materials and methods

We conducted a retrospective, observational, single-center, cohort study with the aim of developing a simple high-accuracy scoring system to predict malignant IPMNs.

### Patients and data collection

We retrospectively analyzed 182 cases of IPMNs who underwent pancreatic resection at Tokyo Medical and Dental University Hospital from January 2008 to June 2024, utilizing medical record entries. Data was collected from January 19, 2023 to August 31, 2024, with strict adherence to not accessing personally identifiable information, both during and after data collection.

Tokyo Medical and Dental University Hospital is one of the leading high-volume centers in Japan, with over 400 IPMN patients referred annually. At our hospital, we adhere to the International Consensus Guidelines for the management of IPMN [6]. We use CT/MRI or EUS to screen for worrisome features (WF) and high-risk stigmata (HRS). HRS include obstructive jaundice, enhancing mural nodules ≥5 mm, and a main pancreatic duct diameter ≥10 mm. WFs include pancreatitis, cysts ≥30 mm, enhancing mural nodules <5 mm, thickened/enhancing cyst walls, main duct size 5–9 mm, abrupt changes in the caliber of the pancreatic duct with distal pancreatic atrophy, lymphadenopathy, increased serum levels of CA-19-9, and cyst growth rate >2.5 mm/year. If HRS is confirmed, surgery is recommended. For cases with confirmed WF, we perform EUS and recommend surgery if enhancing mural nodules ≥5 mm are identified. If no nodules are detected on EUS, we suggest both observation and surgery, with the final decision made after consultation with the patient. The patient has the option to proceed with surgery at any time if they choose.

Considering the aspects highlighted in the previous guidelines or studies, the analysis included the perioperative patient's age, sex, ICG-designated IPMN type, lesion site, pancreatitis history, diabetes mellitus status, presence/absence of jaundice, carcinoembryonic antigen (CEA) and carbohydrate antigen 19–9 (CA19-9) serum levels, ≤6 months preoperative imaging findings, and pathological findings. When assessing the history of pancreatitis, cases with apparent causes, such as alcohol or common bile duct stones, were excluded. In the context of diabetes, the onset of new diabetes was defined as cases where diabetes had not been previously identified and was diagnosed for the first time during the evaluation of IPMN. Additionally, rapid exacerbation was defined as an increase in HbA1c levels of 1% or more within three months. Both new onset and rapid exacerbation were classified as distinct categories, separate from regular diabetes. For serum tumor marker levels, we used cutoff values of 5 ng/mL for CEA and 37 U/mL for CA19-9 to categorize the patients into two groups. Since CA 19–9 levels can be influenced by jaundice, we made sure to include CA 19–9 values measured after the jaundice had been resolved prior to surgery in such cases.

The imaging factors were based on findings associated with malignancy as referenced in the ICG, European EBG, and AGA IG. All imaging findings were reassessed by experts. CT and MRI served as complementary examinations, and if either revealed more advanced findings, the one with the more advanced findings was prioritized. Since the introduction of endoscopic ultrasonography (EUS) at our hospital in April 2018, we also include the findings from EUS examinations, if performed, in the analysis. Considering the ease of use following development of the scoring system, the cutoff values for cyst size, main duct size, and nodule size were adopted from the values provided in the ICG. We excluded cyst growth assessment from this analysis due to its inability to be evaluated in a single examination and the inherent challenge in defining it.

## Image acquisition

For abdominal contrast-enhanced CT of pancreatic cysts, a helical scan mode was employed using a 64- or 96- detector-row CT scanners (Aquilion 64 (Canon Medical Systems, Otawara, Japan), Somatom Edge Plus, or Somatom Force (Siemens Healthcare, Erlangen, Germany)). The contrast agents used were Omnipaque (GE Healthcare, Chicago, IL, USA) or Iomeprol (Bracco-Eisai, Milan, Italy), administered via intravenous injection at a dose of 2 ml per kg of body weight, adjusted according to renal function. The findings were evaluated on contrast-enhanced examinations using the portal phase.

MRI scans were performed using MRI systems with magnetic field strengths of either 1.5 Tesla or 3 Tesla (Signa HDxt/Pioneer (GE Healthcare, Chicago, IL, USA), EXCELART

Vantage (Canon Medical Systems, Otawara, Japan)). For MRCP, patients were asked to orally ingest a negative contrast agent (Bose Del, Meiji Seika Pharma, Tokyo, Japan), and when a contrast agent was required, Gadovist or EOB-Primovist (Bayer, Leverkusen, Germany) was used, administered via intravenous injection at a dose of 0.1 ml per kg of body weight, also adjusted according to renal function.

EUS was performed using the ultrasound gastrovideoscope GF-UCT260 (Olympus, Tokyo, Japan) and the ultrasound processor EU-ME2 PREMIER PLUS (Olympus, Tokyo, Japan). When mural nodules were suspected, the lesion was initially confirmed to be an obvious protruding mass within the cyst from multiple directional views using EUS. Subsequently, an ultrasound contrast agent, Sonazoid (GE Healthcare, Chicago, Illinois, USA), was administered at a dose of 0.015 mL/kg to assess enhancement. Mural nodules that exhibited enhancement were evaluated based on their height relative to the wall. In distinguishing thickened walls, we considered the unevenness of the cyst wall to be a crucial factor, and a thickness of approximately 2 mm or more was deemed thickened if it was notably uneven, as previously reported [20].

## Pathological evaluation

The pathological diagnosis of resected specimens was categorized on the basis of the World Health Organization histological classification of IPMN and the classification from the Baltimore consensus meeting as low-grade IPMN, high-grade IPMN, and intraductal papillary mucinous carcinoma (IPMC) [21, 22]. IPMC was defined as the presence of clear pathological continuity from IPMN to invasive ductal carcinoma. The initial pathological diagnosis was made by three pathologists, and YK further reviewed the IPMC cases to confirm the consistency of the IPMC diagnosis. In this study, low-grade IPMN was classified as benign, whereas high-grade IPMN and IPMC were categorized as malignant.

## Statistical analysis

The median and interquartile range were used to summarize continuous variables, and frequencies and proportions were used to summarize categorical variables. The Mann-Whitney $U$ test and Chi-squared test were used to compare the distributions of continuous and categorical variables between the benign and malignant groups, respectively. For each categorical variable, the odds ratios (ORs) of malignancy and 95% confidence intervals (95% CIs) were also calculated using both univariate and multivariate logistic regression analyses. In conducting the multivariate analysis, IPMN classification was adjusted due to anticipated multicollinearity with the main duct size. Therefore, classifications were based on the presence or absence of cysts, combining the branch duct type and mixed type into a single category while treating the main duct type separately. Additionally, for parameters with small sample sizes, adjustments between groups were made as necessary. Receiver operating characteristic (ROC) analysis was performed to evaluate the variable's ability to diagnose malignant IPMN. The DeLong test was performed to compare the area under the ROC curve (AUC) for each scoring model created by combining factors. Furthermore, we used the model that yielded the highest AUC to calculate each diagnostic performance measure. Statistical significance was set at $p < 0.05$. All statistical analyses were performed in STATA 16.1 software (Stata Corp., TX, USA).

## Developing the scoring system

In developing the scoring system, cases from January 2008 to December 2022 were utilized for its creation, and validation was conducted using cases from January 2023 to June 2024.

The malignancy ORs were calculated for all examined factors using both univariate and multivariate logistic regression analyses. Using ORs directly to assign weights to factors results in complex values that are not suitable for everyday use of the scoring system. To enhance practicality in routine settings, weights were assigned on a scale from 0 to 2. Thus, 1 point was assigned to ORs from 2–5, and 2 points to ORs >5 after discussion with clinical experts in this field (MK, KA, MT, TF, and RO). We created a scoring system that maximizes the AUC through the addition and subtraction of factors. We also assessed the effect of the presence or absence of EUS on the AUC of the scoring system for discriminating malignancy. The suitability of the developed scoring system was verified using cases from January 2023 to June 2024.

### Ethical considerations

This study was conducted with the approval of the Committee on Life Sciences and Bioethics of Tokyo Medical and Dental University Hospital (Permission No. 2000-1080-1, M2019-287-01). As a result, the study was carried out in strict adherence to the ethical standards outlined in the 1964 Declaration of Helsinki and its subsequent amendments. Furthermore, this study involved human participants and was conducted in full compliance with all pertinent guidelines and regulations.

## Results

The clinicopathological characteristics of the 160 cases that underwent resection between January 2008 and December 2022, used in creating the scores, are summarized in Table 1. Out of 160 individuals, 89 (55.6%) had low-grade dysplasia, 32 (20%) had high-grade dysplasia, and 39 (24.4%) had IPMC. There were no significant differences in sex, lesion site, pancreatitis, or CEA levels between the malignant and benign cases. However, significant differences were observed in age, IPMN type, diabetes mellitus status, jaundice, and CA19-9 levels. In particular, jaundice was only observed in the malignant cases, and a distinct association with malignancy was evident. CT was conducted in 154 cases (97.5%), whereas MRI was performed in 135 (84.4%) cases, and at least one of these examinations was performed in all cases. EUS was performed in 55 (34.4%) cases.

The breakdown of imaging findings associated with malignancy in IPMN for each imaging modality is presented in Table 2. In CT/MRI, "enhancing mural nodules," "thickened cyst wall," "main duct size," and "abrupt change in caliber of pancreatic duct with distal pancreatic atrophy" were significantly associated with malignancy. However, "cyst size" and "lymphadenopathy" were not significantly associated with malignancy. Additionally, the significant differences in cyst size and lymphadenopathy detected by EUS are due to the higher number of cases without EUS in the benign group. These significant differences do not have substantive meaning.

In developing the scoring system, Table 3 presents the ORs for factors that demonstrated differences between benign and malignant cases based on clinical and imaging findings. Type of IPMN was adjusted due to anticipated multicollinearity with the main duct size. Consequently, we categorized IPMN based on the presence or absence of cysts, combining branch duct type and mixed type into a single category while treating the main duct type separately. Jaundice was excluded from the analysis, as it was observed only in malignant cases. For mural nodules, multivariate analysis was challenging due to insufficient sample sizes for nodules <5 mm, which resulted in inadequate statistical power and limitations in result interpretation. As a result, we reclassified the nodules into two groups: 'absent or <5 mm' and '≥5 mm'. This was due to the difficulty in definitively identifying nodules smaller than 5 mm. The findings with ORs ranging from 2–5 were age ≥65 years, main duct type, presence of diabetes mellitus,

**Table 1. Characteristics of the IPMN patients in the model development.**

| | N (%) | Benign | Malignant | *p* |
|---|---|---|---|---|
| **Total number** | | **89 (55.6)** | **71 (44.4)** | |
| Age (years) | Median (IQR) | 65 (60–73) | 71 (65–77) | **0.016** |
| <65 years | N (%) | 43 (48.3) | 17 (23.9) | **0.002** |
| ≥65 years | N (%) | 46 (51.7) | 54 (76.1) | |
| Sex | | | | |
| Male | N (%) | 58 (65.2) | 44 (62.0) | 0.898 |
| Female | N (%) | 31 (34.8) | 27 (38.0) | |
| Type of IPMN | | | | |
| Branch duct type | N (%) | 39 (43.8) | 12 (16.9) | **<0.001** |
| Mixed type | N (%) | 46 (51.7) | 51 (71.8) | |
| Main duct type | N (%) | 4 (4.5) | 8 (11.3) | |
| Site of lesion | | | | |
| Head | N (%) | 58 (65.2) | 54 (76.1) | 0.31 |
| Body | N (%) | 20 (22.5) | 10 (14.1) | |
| Tail | N (%) | 11 (12.4) | 7 (9.9) | |
| Pancreatitis | | | | |
| absent | N (%) | 84 (94.4) | 67 (94.4) | >0.999 |
| present | N (%) | 5 (5.6) | 4 (5.6) | |
| Diabetes mellitus | | | | |
| absent | N (%) | 75 (84.3) | 46 (64.8) | **0.007** |
| present | N (%) | 11 (15.7) | 14 (19.7) | |
| new onset / rapid exacerbation | N (%) | 3 (3.4) | 11 (15.5) | |
| Jaundice | | | | |
| absent | N (%) | 89 (100.0) | 62 (87.3) | **<0.001** |
| present | N (%) | 0 (0.0) | 9 (12.7) | |
| CEA (ng/ml) | Median (IQR) | 2.5 (1.4–3.7) | 2.4 (1.9–3.6) | 0.920 |
| <5 ng/ml | N (%) | 77 (86.5) | 62 (87.3) | 0.880 |
| ≥5 ng/ml | N (%) | 12 (13.5) | 9 (12.7) | |
| CA19-9 (U/ml) | Median (IQR) | 10.2 (5.9–17.2) | 11.5 (7.7–25.5) | 0.089 |
| <37 U/ml | N (%) | 88 (98.9) | 58 (81.7) | **<0.001** |
| ≥37 U/ml | N (%) | 1 (1.1) | 13 (18.3) | |
| CT | | | | |
| absent | N (%) | 4 (4.5) | 2 (2.8) | 0.837 |
| present (without contrast) | N (%) | 3 (3.8) | 2 (2.8) | |
| present (with contrast) | N (%) | 82 (92.1) | 67 (94.34) | |
| Time to surgery from CT (days) | Median (IQR) | 49 (27–84) | 43 (20–73) | 0.460 |
| MRI | | | | |
| absent | N (%) | 13 (14.6) | 12 (16.9) | 0.863 |
| present (without contrast) | N (%) | 60 (67.4) | 45 (63.4) | |
| present (with contrast) | N (%) | 16 (18.0) | 14 (19.7) | |
| Time to surgery from MRI (days) | Median (IQR) | 81 (52–114) | 66 (42–100) | 0.100 |
| EUS | | | | |
| absent | N (%) | 66 (74.2) | 39 (54.9) | **0.038** |
| present (without contrast) | N (%) | 8 (9.0) | 12 (16.9) | |
| present (with contrast) | N (%) | 15 (16.9) | 20 (28.2) | |
| Time to surgery from EUS (days) | Median (IQR) | 79 (49–118) | 68 (35–91) | 0.140 |
| Pathology | | | | |

*(Continued)*

**Table 1.** (Continued)

|  |  | Benign | Malignant | *p* |
|---|---|---|---|---|
| **Total number** | **N (%)** | **89 (55.6)** | **71 (44.4)** |  |
| Low-grade dysplasia | N (%) | 89 (100) | 0 (0.0) | **<0.001** |
| High-grade dysplasia | N (%) | 0 (0.0) | 32 (45.1) |  |
| IPMC | N (%) | 0 (0.0) | 39 (54.9) |  |

Statistically significant *p*-values are indicated in bold.

IPMN, intraductal papillary mucinous neoplasm; IQR, interquartile range; CEA, carcinoembryonic antigen; CA19-9, carbohydrate antigen 19–9; CT, computed tomography; MRI, magnetic resonance imaging; EUS, endoscopic ultrasonography; IPMC, intraductal papillary mucinous carcinoma

presence of mural nodule and thickened cyst wall, main duct size of ≥5 mm and <10 mm, and abrupt change in caliber. Findings with ORs ranging from 5 or higher included new onset/rapid exacerbation of diabetes mellitus, CA19-9 of ≥37 U/ml and main duct size of ≥10

**Table 2. Breakdown of imaging findings associated with malignancy in IPMN for each imaging modality.**

|  |  | CT/MRI[†] | | | | EUS | | |
|---|---|---|---|---|---|---|---|---|
|  |  | Benign | Malignant | *p* | | Benign | Malignant | *p* |
|  |  | **89 (55.6)** | **71 (44.4)** | | | **89 (55.6)** | **71 (44.4)** | |
| Cyst size |  |  |  |  | |  |  |  |
| <30 mm | N (%) | 24 (27.0) | 22 (31.0) | 0.577 | | 9 (10.1) | 9 (12.7) | **0.027** |
| ≥30 mm | N (%) | 65 (73.0) | 49 (69.0) |  | | 12 (13.5) | 21 (29.6) |  |
| missing | N (%) | 0 (0.0) | 0 (0.0) |  | | 68 (76.4) | 41 (57.8) |  |
| Enhancing mural nodule |  |  |  |  | |  |  |  |
| absent | N (%) | 67 (75.3) | 37 (52.1) | **0.006** | | 15 (16.9) | 9 (12.7) | **0.003** |
| <5 mm | N (%) | 1 (1.1) | 4 (5.6) |  | | 2 (2.3) | 4 (5.6) |  |
| ≥5 mm | N (%) | 21 (23.6) | 30 (42.3) |  | | 6 (6.7) | 19 (26.8) |  |
| missing | N (%) | 0 (0.0) | 0 (0.0) |  | | 66 (74.2) | 39 (54.9) |  |
| Thickened cyst wall |  |  |  |  | |  |  |  |
| absent | N (%) | 81 (91.0) | 53 (74.7) | **0.005** | | 8 (9.0) | 5 (7.0) | **0.010** |
| present | N (%) | 8 (9.0) | 18 (25.4) |  | | 15 (16.9) | 27 (38.0) |  |
| missing | N (%) | 0 (0.0) | 0 (0.0) |  | | 66 (74.2) | 39 (54.9) |  |
| Main duct size |  |  |  |  | |  |  |  |
| <5 mm | N (%) | 42 (47.2) | 13 (18.3) | **<0.001** | | 9 (10.1) | 8 (11.3) | **0.010** |
| ≥5 mm and <10 mm | N (%) | 34 (38.2) | 32 (45.1) |  | | 12 (13.5) | 13 (18.3) |  |
| ≥10 mm | N (%) | 13 (14.6) | 26 (36.6) |  | | 2 (2.3) | 11 (15.5) |  |
| missing | N (%) | 0 (0.0) | 0 (0.0) |  | | 66 (74.2) | 39 (54.9) |  |
| Abrupt change in caliber of pancreatic duct with distal pancreatic atrophy |  |  |  |  | |  |  |  |
| absent | N (%) | 78 (87.6) | 51 (71.8) | **0.012** | | 20 (22.5) | 26 (36.6) | **0.034** |
| present | N (%) | 11 (12.3) | 20 (28.2) |  | | 3 (3.4) | 6 (8.5) |  |
| missing | N (%) | 0 (0.0) | 0 (0.0) |  | | 66 (74.2) | 39 (54.9) |  |
| Lymphadenopathy |  |  |  |  | |  |  |  |
| absent | N (%) | 87 (97.8) | 68 (95.8) | 0.475 | | 23 (25.8) | 32 (45.1) | **0.011** |
| present | N (%) | 2 (2.3) | 3 (4.2) |  | | 0 (0.0) | 0 (0.0) |  |
| missing | N (%) | 0 (0.0) | 0 (0.0) |  | | 66 (74.2) | 39 (54.9) |  |

Statistically significant p-values are indicated in bold.

†"CT/MRI" indicates the presence of findings if there are findings from either CT or MRI. IPMN, intraductal papillary mucinous neoplasm; CT, computed tomography; MRI, magnetic resonance imaging; EUS, endoscopic ultrasonography

**Table 3. Logistic regression analysis of Odds ratios for each factor.**

| | CT/MRI | | CT/MRI/EUS | |
|---|---|---|---|---|
| | **Univariate** | **Multivariate** | **Univariate** | **Multivariate** |
| | **odds ratio (95%CI)** | **odds ratio (95%CI)** | **odds ratio (95%CI)** | **odds ratio (95%CI)** |
| Age | | | | |
| <65years | 1 | 1 | 1 | 1 |
| >65years | **2.97 (1.50–5.89)** | 1.89 (0.86–4.15) | **2.13 (1.06–4.29)** | **2.27 (1.00–5.16)** |
| Type of IPMN | | | | |
| Branch duct type / Mixed type | 1 | 1 | 1 | 1 |
| Main duct type | 2.69 (0.78–9.36) | 1.34 (0.28–6.43) | 2.69 (0.78–9.36) | 0.96 (0.20–4.64) |
| Diabetes mellitus status | | | | |
| absent | 1 | 1 | 1 | 1 |
| present | 2.08 (0.87–4.96) | 1.94 (0.69–5.46) | 2.08 (0.87–4.96) | 1.73 (0.60–5.03) |
| New onset / Rapid exacerbation | **5.98 (1.58–22.57)** | 3.60 (0.69–18.63) | **5.98 (1.58–22.57)** | 3.62 (0.64–20.64) |
| Jaundice | | | | |
| absent | 1 | | 1 | |
| present | -* | | -* | |
| CA19-9 | | | | |
| <37U/ml | 1 | 1 | 1 | 1 |
| ≥37U/ml | **19.7 (2.51–154.88)** | 7.07 (0.78–64.13) | **19.7 (2.51–154.88)** | 6.98 (0.73–67.13) |
| Enhancing mural nodule | | | | |
| absent / <5mm | 1 | 1 | 1 | 1 |
| ≥5mm | **2.37 (1.20–4.67)** | 2.18 (0.97–4.90) | **3.50 (1.80–6.81)** | **3.34 (1.50–7.42)** |
| Thickened cyst wall | | | | |
| absent | 1 | 1 | 1 | **1** |
| present | **3.43 (1.40–8.47)** | **4.11 (1.44–11.73)** | **3.35 (1.70–6.63)** | **2.47 (1.10–5.51)** |
| Main duct size | | | | |
| <5 mm | 1 | 1 | 1 | 1 |
| ≥5 mm and <10 mm | **3.04 (1.38–6.68)** | **3.42 (1.35–8.63)** | **2.63 (1.18–5.90)** | 2.41 (0.93–6.23) |
| ≥10 mm | **6.46 (2.60–16.07)** | **6.50 (2.07–20.46)** | **7.25 (2.89–18.19)** | **7.08 (2.16–23.14)** |
| Abrupt change in caliber of pancreatic duct with distal pancreatic atrophy | | | | |
| absent | 1 | 1 | 1 | 1 |
| present | **2.78 (1.23–6.29)** | 0.88 (0.30–2.63) | **2.98 (1.32–6.70)** | 1.05 (0.35–3.18) |

Statistically significant *p*-values are indicated in bold.

*All cases with jaundice were malignant and unable to calculate the odds ratio.

†"CT/MRI" indicates the presence of findings if there are findings from either CT or MRI.

‡"CT/MRI/EUS" indicates the presence of findings if there are any findings from CT, MRI, and/or EUS.

IPMN, intraductal papillary mucinous neoplasm; CA19-9, carbohydrate antigen 19–9; CT, computed tomography; MRI, magnetic resonance imaging; EUS, endoscopic ultrasonography; 95% CI, 95% confidence interval

mm. The OR for enhancing mural nodules ≥5mm detected on CT/MRI was 2.37 (95% CI 1.20–4.67), but this increased to 3.50 (95% CI 1.80–6.81) when EUS was also added.

Factors were considered for inclusion in the scoring system by assigning 1 point for an OR between 2 and 5, and 2 points for an OR greater than 5 (Table 4). The mural nodules ≥5mm, thickened wall, and main duct size, which were clearly associated with malignancy in the multivariate analysis, served as the foundation. Other factors were included only if they contributed to an increase in the AUC.

**Table 4. Optimization of factors through comparison of AUC on the basis of the model composition.**

| Model | 1 | 2 | 3 | 4 | 5 | 6 | 7 | 8 | 9 | 10 |
|---|---|---|---|---|---|---|---|---|---|---|
| Age (<65 years, ≥65 years) | | | ● | | | ● | ● | ● | ● | ● |
| Type of IPMN | | | | ● | | | | | | |
| Diabetes mellitus status | | | | | | ● | ● | ● | ● | ● |
| CA19-9 (<37 U/ml, ≥37 U/ml) | | | | | | ● | ● | ● | ● | ● |
| Enhancing mural nodule (absent / <5mm, ≥5mm) | | | | | | | | | | |
| CT/MRI[†] | ● | ● | ● | ● | ● | ● | | ● | ● | |
| CT/MRI/EUS[‡] | | | | | | | ● | | | ● |
| Thickened cyst wall | | | | | | | | | | |
| CT/MRI[†] | ● | ● | ● | ● | ● | ● | ● | | ● | |
| CT/MRI/EUS[‡] | | | | | | | | ● | | ● |
| Main duct size (<5 mm, ≥5 mm and <10 mm, ≥10 mm) | | | | | | | | | | |
| CT/MRI[†] | ● | ● | ● | ● | ● | ● | ● | ● | | |
| CT/MRI/EUS[‡] | | | | | | | | | ● | ● |
| Abrupt change in caliber of pancreatic duct with distal pancreatic atrophy | | | | | | | | | | |
| CT/MRI[†] | | ● | | | | | | | | |
| CT/MRI/EUS[‡] | | | | | | | | | | |
| AUC | 0.736 | 0.733 | 0.759 | 0.737 | 0.759 | 0.781 | 0.797 | 0.790 | 0.788 | 0.807 |
| 95% CI | 0.663–0.809 | 0.658–0.807 | 0.688–0.829 | 0.663–0.810 | 0.687–0.830 | 0.712–0.851 | 0.730–0.864 | 0.723–0.857 | 0.720–0.856 | 0.742–0.872 |
| p-value compared with model 1 | - | 0.815 | 0.218 | 0.877 | 0.292 | 0.111 | | | | |
| p-value compared with model 6 | | | | | | - | **0.048** | 0.459 | 0.347 | 0.157 |

Jaundice was excluded because it was a finding clearly associated with malignancy.

†"CT/MRI" indicates the presence of findings if there are findings from either CT or MRI.

‡"CT/MRI/EUS" indicates the presence of findings if there are any findings from CT, MRI, and/or EUS.

AUC, area under the curve; IPMN, intraductal papillary mucinous neoplasm; CA19-9, carbohydrate antigen 19–9; CT, computed tomography; MRI, magnetic resonance imaging; EUS, endoscopic ultrasonography; 95% CI, 95% confidence interval

An abrupt change in the caliber of the pancreatic duct with distal pancreatic atrophy was excluded from the factors because its inclusion resulted in a decrease in the AUC. Age (<65 years, >65 years), diabetes mellitus status, and CA19-9 (<37 U/ml, >37 U/ml) were included as factors because their addition resulted in an increase in the AUC. However, the type of IPMN did not affect the AUC, so it was excluded for the sake of simplification. The AUC for imaging findings alone was 0.736 (95% CI 0.663–0.809), but it increased to 0.781 (95% CI 0.712–0.851) with the inclusion of clinical findings (Table 4 and Fig 1).

Ultimately, the IPMN Malignancy Prediction (IMAP) scoring system was established with six factors: imaging findings including mural nodules, thickened wall, and main duct size, along with clinical factors such as age, diabetes status, and CA19-9 levels. (Table 5).

We performed a comparison to assess the additional effect of including EUS evaluation for each imaging finding on the AUC of the IMAP scores. The AUC was 0.790 (95% CI 0.723–0.857) for Model 8 when thickened cyst wall was evaluated by EUS and was 0.788 (95% CI 0.720–0.856) for Model 9 when main duct size was evaluated by EUS. Both models increased the AUC. However, when EUS was used to evaluate the enhancing mural nodules, Model 7 gave the highest AUC among the three models, which achieved a value of 0.797 (95% CI 0.730–0.864). Model 10, wherein all imaging factors were evaluated by EUS,

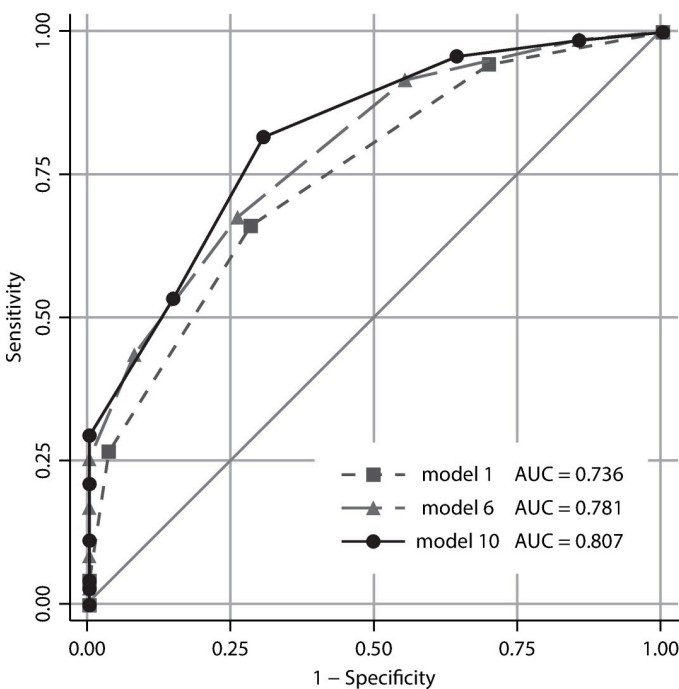

**Fig 1. ROC curves for each malignancy prediction model.** ROC curve, receiver operating characteristic curve; AUC, area under the ROC curve.

provided the highest AUC of 0.807 (95% CI 0.742–0.872) (Fig 1). When using Model 10 to calculate the diagnostic performance measures of the IMAP scoring system, a score of ≥5 resulted in 100% specificity and a positive predictive value (Tables 6 and S1). For a cutoff value of 3, the sensitivity and specificity were 78.9% (95% CI 67.6–87.7) and 69.7% (95% CI 59.0–79.0), respectively. The positive predictive value was 67.5% (95% CI 56.3–77.4), indicating that 7 out of 10 positive cases were predicted to be malignant.

The suitability of the developed scoring system was evaluated using 22 cases from January 2023 to June 2024 (S2 Table). In the validation analysis, a score of 6 or more achieved 100% specificity and positive predictive value. Using a cutoff value of 3 points in validation yielded a sensitivity of 83.3% (95% CI 35.9–99.6), which was comparable to the sensitivity during the

**Table 5. IMAP scoring system.**

| Score | 0 | 1 | 2 |
|---|---|---|---|
| Age (years) | <65 | ≥65 | |
| Diabetes mellitus status | - | + | New onset Rapid exacerbation |
| CA19-9 (U/ml) | <37 | | ≥37 |
| Enhancing mural nodule (mm) | absent / <5mm | ≥5mm | |
| Thickened cyst wall | - | + | |
| Main duct size (mm) | <5 | 5–10 | ≥10 |

Jaundice was excluded because it was a finding clearly associated with malignancy.

IMAP, IPMN malignancy prediction; IPMN, intraductal papillary mucinous neoplasm; CA19-9, carbohydrate antigen 19–9

**Table 6. Diagnostic performance measures based on the total IMAP score using model 10.**

| Total score | Sensitivity (%) (95% CI) | Specificity (%) (95% CI) | Positive predictive value (%) (95% CI) | Negative predictive value (%) (95% CI) |
|---|---|---|---|---|
| 1 | 98.6 (92.4–100) | 14.6 (8.01–23.7) | 47.9 (39.6–56.4) | 92.9 (66.1–99.8) |
| 2 | 94.4 (86.2–98.4) | 37.1 (27.1–48.0) | 54.5 (45.2–63.5) | 89.2 (74.6–97.0) |
| 3 | 78.9 (67.6–87.7) | 69.7 (59.0–79.0) | 67.5 (56.3–77.4) | 80.5 (69.9–88.7) |
| 4 | 50.7 (38.6–62.8) | 86.5 (77.6–92.8) | 75.0 (60.4–86.4) | 68.8 (59.3–77.2) |
| 5 | 29.6 (19.3–41.6) | 100 (95.9–100) | 100 (83.9–100) | 64.0 (55.5–72.0) |
| 6 | 21.1 (12.3–32.4) | 100 (95.9–100) | 100 (78.2–100) | 61.4 (52.9–69.3) |

IMAP, IPMN malignancy prediction; IPMN, intraductal papillary mucinous neoplasm; 95% CI, 95% confidence interval

scoring development. However, specificity decreased to 37.5% (95% CI 15.2–64.6) from 69.7% (95% CI 59.0–79.0) observed during the scoring development (Tables 7 and S3).

## Discussion

In this retrospective analysis of IPMN surgical cases at our institution, when classifying low-grade dysplasia as benign and high-grade dysplasia and IPMC as malignant, significant differences were observed for different combinations of age, diabetes mellitus status, jaundice, and CA19-9 levels in distinguishing between benign and malignant cases. Additionally, significant differences were noted in the imaging findings, including enhancing mural nodules ≥5mm, thickened cyst wall, main duct size, and abrupt changes in caliber of the pancreatic duct with distal atrophy. We identified six significant factors for inclusion in the IMAP scoring system: age, diabetes mellitus status, CA19-9 level, enhancing mural nodule ≥5mm, thickened cyst wall, and main duct size. All of these factors are familiar parameters and easily measurable during noninvasive examinations. The IMAP scoring system, despite being simpler than the previous scoring system, achieved similar performance, with an AUC of 0.781 (95% CI 0.712–0.851) 0.781 for discriminating malignancy [10–19]. Furthermore, when incorporating EUS for image assessment, the AUC reached 0.807 (95% CI 0.742–0.872), surpassing the performance of the previous scoring system [11, 12, 15, 17].

EUS has a higher resolution than both MRI and CT, providing more detailed images of pancreatic cystic lesions [23]. In the recently revised ICG guideline, there is acknowledgment of the potential usefulness of EUS for distinguishing between benign and malignant IPMN [6]. However, the European EBG notes that the discriminatory ability of EUS between benign and malignant lesions is controversial [5], while the AGA IG suggests MRI for follow-up and considers EUS as an adjunct [4]. Although the effect of operator skill on imaging findings has

**Table 7. Validation using IMAP score model 10.**

| Total score | Sensitivity (%) (95% CI) | Specificity (%) (95% CI) | Positive predictive value (%) (95% CI) | Negative predictive value (%) (95% CI) |
|---|---|---|---|---|
| 1 | - | - | - | - |
| 2 | 100 (54.1–100) | 6.25 (0.2–30.2) | 28.6 (11.3–52.2) | 100 (2.5–100) |
| 3 | 83.3 (35.9–99.6) | 37.5 (15.2–64.6) | 33.3 (11.8–61.6) | 85.7 (42.1–99.6) |
| 4 | 50.0 (11.8–88.2) | 81.3 (54.4–96.0) | 50.0 (11.8–88.2) | 81.3 (54.4–96.0) |
| 5 | 33.3 (4.3–77.7) | 93.8 (69.8–99.8) | 66.7 (9.4–99.2) | 78.9 (54.4–93.9) |
| 6 | 16.7 (0.4–64.1) | 100 (79.4–100) | 100 (2.5–100) | 76.2 (52.8–91.8) |

IMAP, IPMN malignancy prediction; IPMN, intraductal papillary mucinous neoplasm; 95% CI, 95% confidence interval

been considered significant in EUS, making it less amenable to standardization, it is worth noting that at our institution, despite a relatively short period since its introduction and a limited utilization rate of 34.4% for EUS, we have successfully and conclusively demonstrated the effectiveness of EUS for diagnosis of malignancies. Specifically, as previously reported, nodule detection by EUS has been shown to make the most significant contribution to the evaluation of malignancy [24–26]. In this study, nodules smaller than 5 mm, which are challenging to assess, were excluded from evaluation. Despite this exclusion, the ability to assess malignancy significantly improved, with the AUC increasing from 0.781 to 0.797. Given the minimally invasive nature of EUS and its augmenting role in malignancy diagnosis, it should be strongly considered for IPMNs.

In this study, the presence and size of cysts were not significantly associated with malignancy. Consistent with previous reports, main-duct IPMN tended to show a higher risk of malignancy; however, in multivariate analysis, the main-duct type itself was not a significant factor. Instead, the diameter of the main pancreatic duct was found to be a more critical factor in assessing the risk of malignancy. Several studies have emphasized the importance of cyst size in malignant evaluation, but others have suggested a limited association [7, 8, 18]. In clinical practice, the method for measuring cyst size is often ambiguous, especially when there are multiple adjacent cysts or when these cysts enlarge enough to eliminate the interstitial space. The IMAP scoring system, which has a strong ability to differentiate high malignancies from other degrees without needing to evaluate the cyst diameter, can be described as a user-friendly system in real clinical practice. To simplify the scoring system, we omitted cyst growth as a factor, given the difficulty in accurately assessing it. However, we believe that in the future, the association between cyst growth and malignancy should be evaluated using clearly defined criteria and precise measurements.

This study had some limitations. Foremost, Since this study is based on a retrospective analysis of surgical cases, a prospective study involving a larger number of cases is needed to determine whether the IMAP scoring system can be applied to original preoperative cases. Furthermore, given that this study was conducted at a single university hospital renowned for its high quantity and quality, it is probable that the examinations were accurate, but this degree of accuracy may not possible in other institutions. A multicenter study would be needed to determine if the IMAP scoring system can be used successfully in other facilities.

In this study, we also explored the utility of EUS for diagnosing malignancy in IPMN. However, the number of cases in which EUS could be performed was limited. Given the improvement in malignancy diagnosis through EUS imaging evaluation even with the limited number of cases, the utility of EUS potentially can be further enhanced as more cases are accumulated in the future. EUS is minimally invasive and provides very detailed imaging for pancreatic cystic tumors. Consequently, its utility is promising, and in the future, the accuracy of diagnosing a malignancy may be further enhanced through EUS evaluation.

In diagnosing IPMC, we defined IPMC in this study as cases where continuity from IPMN to invasive ductal carcinoma was pathologically confirmed, thereby minimizing the inclusion of sporadic invasive pancreatic ductal carcinoma. However, it remains challenging to completely differentiate between invasive pancreatic ductal carcinoma originating from IPMN and those coincidentally colliding with IPMN. Recent attempts have been made to distinguish between these two entities based on mutations in genes such as *KRAS*, *GNAS*, *CDKN2A/p16*, *SMAD4*, and *TP53* [27]. Nevertheless, it is still difficult to achieve a complete differentiation using these methods. It is impossible to completely exclude sporadic invasive ductal carcinoma within this study.

While not performed at our institution, diagnostic techniques, such as cytology of pancreatic fluid during endoscopic retrograde cholangiopancreatography (ERCP) and tissue

sampling via EUS-fine-needle aspiration (FNA), are known to enhance diagnostic sensitivity and specificity [28–30]. However, these procedures are all invasive, so it is not realistically feasible to perform them in all cases. Careful consideration is necessary for cases that require these procedures, similar to surgical cases. Although using the imaging findings specified in the guidelines alone can make it challenging to identify malignant cases [7–9], there are many other scoring systems [10–19]. However, these systems often involve complex equations or incorporate invasive procedures to enhance diagnostic accuracy, making them not easily accessible for straightforward use. There is a clinical need for a simple scoring system that relies solely on noninvasive information and provides sufficient justification for invasive procedures. Our goal was to develop such a scoring system.

The IMAP scoring system developed in this study is based on data from cases where surgery was performed for IPMN and is well-suited for cases being considered for surgery. It is composed solely of medical history, blood tests, and imaging studies, allowing for a straightforward assessment of the risk of malignant IPMN. This scoring system enables quick and straightforward decisions regarding whether or not to proceed with surgery immediately, conduct further invasive examinations, or opt for a watch-and-wait approach depending on the assessed risk. This scoring system could be used in a three-tiered classification, similar to the Child-Pugh classification, with 1–2 points indicating group A, 3–5 points indicating group B, and $\geq 6$ points or the presence of jaundice indicating group C, for example. The proposed scoring system could be used to determine the management approach, such as observation for group A, additional evaluation through ERCP or EUS-FNA for group B, and consideration of surgery for group C.

The IMAP scoring system, which incorporates EUS evaluation, is expected to enable selection of the most appropriate treatment for patients without imposing excessive invasiveness.

## Supporting information

**S1 Table. Distribution of IMAP score and malignancy.**
(DOCX)

**S2 Table. Characteristics of the IPMN patients in the validation cases.**
(DOCX)

**S3 Table. Distribution of IMAP score and malignancy in the validation cases.**
(DOCX)

**S1 Data. All data utilized for the analysis in this article.**
(XLSX)

## Acknowledgments

We would like to thank Ami Kawamoto (Department of Gastroenterology and Hepatology, Tokyo Medical and Dental University) for English correction. We express our gratitude to Dr. Hiroaki Ono, Dr. Hiroki Ueda, and Dr. Shuichi Watanabe (Department of Hepato-Biliary-Pancreatic Surgery, Tokyo Medical and Dental University) for their support.

## Author Contributions

**Conceptualization:** Masanori Kobayashi, Keiichi Akahoshi, Takeo Fujiwara, Minoru Tanabe, Ryuichi Okamoto.

**Data curation:** Masanori Kobayashi, Hiromune Katsuda, Aya Maekawa, Keiichi Akahoshi, Ryosuke Watanabe, Yuko Kinowaki.

**Formal analysis:** Masanori Kobayashi, Hisaaki Nishimura, Takeo Fujiwara.

**Investigation:** Hiromune Katsuda, Aya Maekawa, Keiichi Akahoshi, Ryosuke Watanabe, Yuko Kinowaki.

**Validation:** Masanori Kobayashi, Hiromune Katsuda, Aya Maekawa, Keiichi Akahoshi, Ryosuke Watanabe, Yuko Kinowaki, Takeo Fujiwara, Minoru Tanabe, Ryuichi Okamoto.

**Writing – original draft:** Masanori Kobayashi, Hisaaki Nishimura.

**Writing – review & editing:** Masanori Kobayashi, Takeo Fujiwara, Minoru Tanabe, Ryuichi Okamoto.

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
