## [Decision Letter · Decision Letter 0]

30 Jul 2024

PONE-D-24-16942Development of an intraductal papillary mucinous neoplasm malignancy prediction scoring systemPLOS ONE

Dear Dr. Kobayashi,

Thank you for submitting your manuscript to PLOS ONE. After careful consideration, we feel that it has merit but does not fully meet PLOS ONE’s publication criteria as it currently stands. Therefore, we invite you to submit a revised version of the manuscript that addresses the points raised during the review process.

I am grateful to the authors for the opportunity to evaluate their study. The topic is interesting and relevant to clinical practice. I think the reviewers have done a good job and provided insights to try to improve the paper.  

I particularly point out:

Reviewer 1 requires further investigation regarding score validation. The definitions requested in item 2 are relevant (new-onset diabetes and mural nodules (Reviewer 2 also requests them).  Even if reference is made to definitions found in the ICGs, providing the reader with a quick reference without forcing him or her to retrieve the bibliographic source makes it easier to read the paper.

Reviewer 2: I think points 2, 4, deserve a detailed reply.

Reviewer 4 I think it is essential to find an appropriate answer to notations 1 and 2

I also add my personal comments: is the Ca 19-9 value also considered in patients with jaundice? Because jaundice has clearly been associated with neoplasia, but it should not be forgotten that it is also associated with an increased Ca 19-9 value. Therefore, it is not clear whether the Ca 19-9 value should also be considered in jaundiced patients. If the answer was yes the author has to explain why (it might introduce a bias in the score, indirectly reporting jaundice within it) ? if the answer was no the author would have to give an adequate explanation in the text.

Parameters already used to direct the choice by surgeons with respect to the dilemma to operate or not to operate ( i.e., the type of IPMN and the presence of HRS and WF) are evaluated in the score.

The clinical utility of a new score should be to help the surgeon in the decision to refer ( or not ) a patient with IPMN to pancreasectomy. The goal of the score for the authors is to be predictive about the finding of neoplasia in the surgical specimen ( and only indirectly to guide surgeons' choice): why should I use it? What does it actually add in the decision-making pathway?

I am aware that the work ahead for the authors is very challenging, but I am convinced that substantial editing of the paper is crucial.

We look forward to receiving your revised manuscript.

Kind regards,

Fabrizio D'Acapito, Ph.D,M.D.

Academic Editor

PLOS ONE

Journal Requirements:

**Additional Editor Comments:**

I am grateful to the authors for the opportunity to evaluate their study. The topic is interesting and relevant to clinical practice. I think the reviewers have done a good job and provided insights to try to improve the paper.

I particularly point out:

Reviewer 1 requires further investigation regarding score validation. The definitions requested in item 2 are relevant (new-onset diabetes and mural nodules (Reviewer 2 also requests them). Even if reference is made to definitions found in the ICGs, providing the reader with a quick reference without forcing him or her to retrieve the bibliographic source makes it easier to read the paper.

Reviewer 2: I think points 2, 4, deserve a detailed reply.

Reviewer 4 I think it is essential to find an appropriate answer to notations 1 and 2

I also add my personal comments: is the Ca 19-9 value also considered in patients with jaundice? Because jaundice has clearly been associated with neoplasia, but it should not be forgotten that it is also associated with an increased Ca 19-9 value. Therefore, it is not clear whether the Ca 19-9 value should also be considered in jaundiced patients. If the answer was yes the author has to explain why (it might introduce a bias in the score, indirectly reporting jaundice within it) ? if the answer was no the author would have to give an adequate explanation in the text.

Parameters already used to direct the choice by surgeons with respect to the dilemma to operate or not to operate ( i.e., the type of IPMN and the presence of HRS and WF) are evaluated in the score.

The clinical utility of a new score should be to help the surgeon in the decision to refer ( or not ) a patient with IPMN to pancreasectomy. The goal of the score for the authors is to be predictive about the finding of neoplasia in the surgical specimen ( and only indirectly to guide surgeons' choice): why should I use it? What does it actually add in the decision-making pathway?

I am aware that the work ahead for the authors is very challenging, but I am convinced that substantial editing of the paper is crucial.

Reviewers' comments:

Reviewer's Responses to Questions

**Comments to the Author**

1. Is the manuscript technically sound, and do the data support the conclusions?

Reviewer #1: Yes

Reviewer #2: Partly

Reviewer #3: Yes

Reviewer #4: No

2. Has the statistical analysis been performed appropriately and rigorously? 

Reviewer #1: Yes

Reviewer #2: Yes

Reviewer #3: Yes

Reviewer #4: No

3. Have the authors made all data underlying the findings in their manuscript fully available?

Reviewer #1: Yes

Reviewer #2: No

Reviewer #3: Yes

Reviewer #4: Yes

4. Is the manuscript presented in an intelligible fashion and written in standard English?

Reviewer #1: Yes

Reviewer #2: Yes

Reviewer #3: Yes

Reviewer #4: Yes

5. Review Comments to the Author

Reviewer #1: This study showed a scoring system for intraductal papillary mucinous neoplasm malignancy prediction. This study may be useful for intraductal papillary mucinous neoplasm management. I have several recommendations.

1.It would be better to validated your scoring system in an independent population. Or, you can divided your patients into training group and test group.

2.How to define new-onset diabetes? How to define mural nodules (contrast enhanced?)?

3.For logistic regression analysis, did you use multivariable logistic analysis or only use univariable logistic analysis?

4.How many patients with 1-8 IMAP score? It would be better to show the patient number with different IMAP score divided by benign and malignant IPMN.

5.Only ORs were considered in IMAP scoring system. ORs were related to sample size. CA19-9 > 37 had a 3 point should be reconsidered.

Reviewer #2: For author

This article is a retrospective study about an intraductal papillary mucinous neoplasm malignancy prediction scoring system. The report is fascinating. however, this report is fascinating; however, several points need to be modified.

Major

1. Regarding Table 1, the definition of rapid exacerbation in diabetes is not provided. Please clarify the definition.

2. In this study, the diagnostic rate varies significantly depending on the type of CT and MRI machines used. Additionally, whether contrast is used or not is also important. Please add these details to the method section.

3. How was the enhancing mural nodule evaluated using EUS? If contrast-enhanced EUS was performed, details about the type of contrast agent and the method should be added.

4. How were thickened cyst wall and mural nodules of 5 mm or less distinguished? Please clarify the definitions.

5. An abrupt change in the caliber of the pancreatic duct with distal pancreatic atrophy is a typical finding suggestive of conventional pancreatic cancer. In the surgical specimens of this study, do they include not only invasive carcinoma derived from IPMN but also conventional pancreatic cancer?

6. Regarding Table 1, a significant difference was reported with EUS in the lymphadenopathy category, Is this correct?

7. How was the type of IPMN classified in preoperative examinations? If the classification is based on the main pancreatic duct diameter, then both the IPMN type and the main pancreatic duct diameter are included as the same indicator in the score. I recommend to include only one of them in the score.

Reviewer #3: The issue reports a very interesting new approach for IPMN scoring model. Even if it was based on a retrospective analysis, either the method or the results seem to be enchouraging to have an improvement of better relation between preoperative and pathological finndings. In particular IMAP model score 6 showed strong specificity and acceptable sensitivity. I think this study should be considered a very concrete step forward in order to assess a useful and practice model in management of IPMN likelihood risk staging.

Reviewer #4: The authors describe a single institutional retrospective study analyzing IPMN cases focusing on the presence of malignancy in those tumors. Also, the authors developed the scoring system to predict the presence of malignancy in IPMN cases (IMAP scoring system). Their scoring system includes Age, IPMN type, DM status, CA19-9, enhancing mural nodule, thickened cyst wall, MPD size, and jaundice. The authors evaluated the significance of IMAP scoring system by calculating AUC, and other parameters (e.g., sensitivity, specificity, positive predictive value, and negative predictive value). Finally, the authors concluded that IMAP scoring system was a clinically useful tool for determining the treatment approach for IPMN.

This manuscript was well written, and the figures and tables were appropriately prepared. The topic of this study is important in the field of the management of IPMN, and the results of this study were not surprising and seemed reasonable. However, this study also contained several critical issues that greatly diminish its importance.

1. First, the most critical issue with this study was that this study included only resected cases, as mentioned by the authors. The authors mentioned that "more than 400 IPMN patients were referred to their institute annually", and thus the authors should have evaluated more than 5600 IPMNs during the study period (2008 - 2022). However, this study included only 160 IPMNs who underwent surgery (less than 3% of their IPMN cohort). In principle, it is essential to include all cases to evaluate the sensitivity and specificity of a given prediction system. Otherwise, the authors should clearly state how surgical eligibility was determined, and surgical eligibility should be consistent throughout the study period.

2. In relation to the issue mentioned above, the prevalence of malignant IPMN (high-grade IPMN and IPMC) is quite important to evaluate the clinical usefulness of prediction system. As mentioned above, the authors should have evaluated 5600 IPMNs during the study period and only 71 cases (1.3%) were determined as malignant IPMN. On the other hand, the analysis in this study was conducted under the prevalence of malignant IPMN of 44%. Because the prevalence of malignant IPMNs significantly impacts on the PPV and NPV, the values of PPV and NPV in this study is inappropriate. Also, IMAP scores in the cohort excluded from this study (more than 5400 cases) should have significant impact on the sensitivity and specificity of IMAP scoring system. In this regard, the values of sensitivity and specificity calculated in this study are quite difficult to interpret. The authors need to consult the statistician regarding the validity of evaluating the IMAP scoring system using the rather limited subset of the cohort.

3. The authors should evaluate the clinical utility of the IMAP scoring system in comparison to other well-established grading systems (e.g., ICG, European EBG, AGA IG).

4. The authors described only discovery cohort regarding the IMAP scoring system. However, a validation cohort is essential to appropriately evaluate the usefulness of this system.

6. PLOS authors have the option to publish the peer review history of their article (what does this mean?). If published, this will include your full peer review and any attached files.

Reviewer #1: No

Reviewer #2: No

Reviewer #3: **Yes: **Andrea Gardini

Reviewer #4: No

---

## [Author Response · Author response to Decision Letter 0]

4 Sep 2024

Point-by-point responses to the comments raised by the editors and reviewers.

We express our deep gratitude for the constructive comments and suggestions from the editor and reviewers. We have made revisions in the manuscript, including clarification of the text. We believe that our revised manuscript has been substantially improved.

Reviewer #1: 

This study showed a scoring system for intraductal papillary mucinous neoplasm malignancy prediction. This study may be useful for intraductal papillary mucinous neoplasm management. I have several recommendations.

Response

Thank you for your review. We greatly appreciate your interest in our study.

1.It would be better to validated your scoring system in an independent population. Or, you can divided your patients into training group and test group.

Response

Thank you for your valuable comments. We have added cases from January 2023 to June 2024 and conducted validation (Table 7, Sp Table 2, Sp Table 3). Although the number of cases was small, we were still able to evaluate malignancy with 100% specificity in the group with an IMAP score of 6 or higher. The IMAP score is intended to be easily used in clinical practice. Although further accumulation of cases is necessary, we believe that the IMAP score is useful for primary screening, and dividing patients into three groups (1-2 points, 3-5 points, and 6 or more points), as discussed in the Discussion section, will help determine the need for invasive procedures.

2.How to define new-onset diabetes? How to define mural nodules (contrast enhanced?)?

Response

Thank you for your valuable comments. We defined new-onset diabetes as cases where diabetes was first diagnosed during the evaluation of IPMN. Additionally, nodules were defined according to the guidelines, where a nodule is one that is enhanced using a contrast agent. We have added this information to the Methods section.

P9 line 122-125

“In the context of diabetes, the onset of new diabetes was defined as cases where diabetes had not been previously identified and was diagnosed for the first time during the evaluation of IPMN. Additionally, rapid exacerbation was defined as an increase in HbA1c levels of 1% or more within three months.”

P11 line 159-162

“When mural nodules were suspected, the ultrasound contrast agent Sonazoid (GE Healthcare, Chicago, IL, USA) was administered at a dose of 0.015 mL/kg to verify enhancement. Enhancing mural nodules were evaluated based on their height from the wall.”

3.For logistic regression analysis, did you use multivariable logistic analysis or only use univariable logistic analysis?

Response

Thank you very much for your important feedback. To enhance the accuracy of the score, we have incorporated multivariate analysis. As noted by Reviewer 2, the type of IPMN exhibits multicollinearity with the main duct size. When we categorized patients into two groups—branch-duct + mixed type and main duct type—neither univariate nor multivariate analysis revealed a significant difference. Additionally, including IPMN type in the score did not improve the AUC, so we have decided to exclude it from the scoring system. Regarding CA19-9, as mentioned in comment 5, although the multivariate analysis did not detect a significant difference, its inclusion did enhance the accuracy of the score. Based on your suggestions, we have assigned 2 points for an odds ratio greater than 5 in the revised scoring system. For enhancing mural nodules, multivariate analysis was challenging due to the insufficient sample sizes for mural nodules <5 mm, resulting in inadequate statistical power and limitations in result interpretation. Therefore, we reclassified the nodules into two groups: 'absent or <5 mm' and '>5 mm. This was due to the difficulty in definitively identifying nodules smaller than 5 mm.

We believe that the addition of multivariate analysis has led to a more accurate scoring system. Thank you for your valuable comment.

4.How many patients with 1-8 IMAP score? It would be better to show the patient number with different IMAP score divided by benign and malignant IPMN.

Response

Thank you for your comments. We have presented the distribution of the IMAP Score and malignancy in Supplement Table 1. As demonstrated in the manuscript, the distribution of benign and malignant cases reverses at a score of 3 points, and only malignant cases are observed at 5 points or higher. We are confident that this demonstrates the development of a highly accurate scoring system.

5.Only ORs were considered in IMAP scoring system. ORs were related to sample size. CA19-9 > 37 had a 3 point should be reconsidered.

Response

Thank you for your comments. As noted in point 3, the multivariate analysis indeed showed that the significant difference observed in the univariate analysis for CA19-9 was lost, likely due to sample size issues. However, it is clear from the univariate analysis that CA19-9, with an odds ratio of 19.7, is strongly correlated with malignancy. In the revised scoring system, we have established a rule to assign 2 points for an odds ratio of 5 or higher. Accordingly, CA19-9 has been assigned 2 points. The inclusion of CA19-9 has led to an improvement in score accuracy, resulting in a more robust scoring system. Thank you for your valuable feedback.

Reviewer #2: 

This article is a retrospective study about an intraductal papillary mucinous neoplasm malignancy prediction scoring system. The report is fascinating. however, this report is fascinating; however, several points need to be modified.

Response

Thank you for your review. We greatly appreciate your interest in our study.

1. Regarding Table 1, the definition of rapid exacerbation in diabetes is not provided. Please clarify the definition.

Response

Thank you for your feedback. We defined a rapid exacerbation in diabetes as an increase in HbA1c of more than 1% within a 3-month period. In Japan, HbA1c is regularly measured in diabetes patients. We have added the definition of rapid exacerbation to the Methods section. Thank you for your comment.

P9 line 124-125

“Additionally, rapid exacerbation was defined as an increase in HbA1c levels of 1% or more within three months.”

2. In this study, the diagnostic rate varies significantly depending on the type of CT and MRI machines used. Additionally, whether contrast is used or not is also important. Please add these details to the method section.

Response

Thank you for your comment. CT scans were performed using either 64-detector or 96-detector CT scanners (Aquilion 64 (Canon Medical Systems, Otawara, Japan), Somatom Edge Plus or Somatom Force (Siemens Healthineers, Erlangen, Germany)). Contrast agents used included Omnipaque (GE Healthcare, Chicago, Illinois, USA) or Iomeprol (Bracco-Eisai, Milan, Italy). MRI was conducted with either 1.5 Tesla or 3 Tesla MRI machines (Signa HDxt/Pioneer (GE Healthcare, Chicago, Illinois, USA), EXCELART Vantage (Canon Medical Systems, Otawara, Japan)). For MRCP, a negative contrast agent (Bosedel, Meiji Seika pharma, Tokyo Japan) was administered orally. When contrast was required, Gadovist or EOB-Primovist (Bayer, Leverkusen, Germany) was used. We have added these details under 'Image Acquisition' in the Methods section and have also included information about the use of contrast agents in Table 1.

 P10 line 140-154

“For abdominal contrast-enhanced CT of pancreatic cysts, a helical scan mode was employed using a 64- or 96- detector-row CT scanners (Aquilion 64 (Canon Medical Systems, Otawara, Japan), Somatom Edge Plus, or Somatom Force (Siemens Healthcare, Erlangen, Germany)). The contrast agents used were Omnipaque (GE Healthcare, Chicago, IL, USA) or Iomeprol (Bracco-Eisai, Milan, Italy), administered via intravenous injection at a dose of 2 ml per kg of body weight, adjusted according to renal function. The findings were evaluated on contrast-enhanced examinations using the portal phase.

MRI scans were performed using MRI systems with magnetic field strengths of either 1.5 Tesla or 3 Tesla (Signa HDxt/Pioneer (GE Healthcare, Chicago, IL, USA), EXCELART Vantage (Canon Medical Systems, Otawara, Japan)). For MRCP, patients were asked to orally ingest a negative contrast agent (Bose Del, Meiji Seika Pharma, Tokyo, Japan), and when a contrast agent was required, Gadovist or EOB-Primovist (Bayer, Leverkusen, Germany) was used, administered via intravenous injection at a dose of 0.1 ml per kg of body weight, also adjusted according to renal function.”

3. How was the enhancing mural nodule evaluated using EUS? If contrast-enhanced EUS was performed, details about the type of contrast agent and the method should be added.

Response

Thank you for your comment. When mural nodules were suspected, we first confirmed the lesion as an obvious protruding mass within the cyst by observing it from multiple directions using EUS. Subsequently, an ultrasound contrast agent, Sonazoid (GE Healthcare, Chicago, Illinois, USA), was administered at a dose of 0.015 mL/kg to assess enhancement. Mural nodules that showed enhancement were evaluated based on their height relative to the wall. This information has been added to the Methods section under 'Image Acquisition.' Thank you for your input.

 P11 Line 159-164

“When mural nodules were suspected, the lesion was initially confirmed to be an obvious protruding mass within the cyst from multiple directional views using EUS. Subsequently, an ultrasound contrast agent, Sonazoid (GE Healthcare, Chicago, Illinois, USA), was administered at a dose of 0.015 mL/kg to assess enhancement. Mural nodules that exhibited enhancement were evaluated based on their height relative to the wall.”

4. How were thickened cyst wall and mural nodules of 5 mm or less distinguished? Please clarify the definitions.

Response

Thank you for your comment. We consider it a very important point. At our institution, as previously reported, wall thickening is assessed based on the unevenness of the cyst wall, with a thickness of approximately 2 mm or more being considered thickened if the unevenness is pronounced (Kobayashi M et al. Pancreas. 53, e521-527. 2024). When a nodule is suspected, we first confirm that the lesion is a clear protrusion into the cyst from multiple angles. If the lesion is confirmed as a protruding mass into the cyst lumen from multiple views and shows contrast enhancement, it is classified as a mural nodule. However, in practice, it is often challenging to differentiate between thickened walls and mural nodules when the nodules are small. Therefore, in developing the scoring system, only nodules measuring 5 mm or larger were included as significant mural nodules. We have added a note in the Results section indicating that only nodules >5 mm were used in the scoring.

 P19 Line 258-263

 “For mural nodules, multivariate analysis was challenging due to insufficient sample sizes for nodules <5 mm, which resulted in inadequate statistical power and limitations in result interpretation. As a result, we reclassified the nodules into two groups: 'absent or <5 mm' and '>5 mm'. This was due to the difficulty in definitively identifying nodules smaller than 5 mm.”

5. An abrupt change in the caliber of the pancreatic duct with distal pancreatic atrophy is a typical finding suggestive of conventional pancreatic cancer. In the surgical specimens of this study, do they include not only invasive carcinoma derived from IPMN but also conventional pancreatic cancer?

Response

Thank you for your comment. The possibility of including sporadic invasive ductal carcinoma in this study is indeed an important consideration. As described in the Methods section, we defined IPMC as cases where continuity from IPMN to invasive ductal carcinoma was clearly established pathologically, in order to minimize the inclusion of sporadic invasive ductal carcinoma. However, it is difficult to fully differentiate between invasive ductal carcinoma originating from IPMN and those coincidentally colliding with IPMN. Although recent attempts have been made to distinguish between invasive ductal carcinoma arising from IPMN and sporadic invasive ductal carcinoma based on mutations in genes such as KRAS, GNAS, CDKN2A/p16, SMAD4, and TP53, these methods do not allow for complete differentiation. We acknowledge the limitations of this definition. An abrupt change in the caliber of the pancreatic duct with distal pancreatic atrophy can occur in IPMN-derived invasive ductal carcinoma, and this finding alone does not conclusively indicate sporadic invasive ductal carcinoma. We have included this discussion in the limitations of the Discussion section.

 P28 Line 377-385

 “In diagnosing IPMC, we defined IPMC in this study as cases where continuity from IPMN to invasive ductal carcinoma was pathologically confirmed, thereby minimizing the inclusion of sporadic invasive pancreatic ductal carcinoma. However, it remains challenging to completely differentiate between invasive pancreatic ductal carcinoma originating from IPMN and those coincidentally colliding with IPMN. Recent attempts have been made to distinguish between these two entities based on mutations in genes such as KRAS, GNAS, CDKN2A/p16, SMAD4, and TP53[27]. Nevertheless, it is still difficult to achieve a complete differentiation using these methods. It is impossible to completely exclude sporadic invasive ductal carcinoma within this study.”

6. Regarding Table 2, a significant difference was reported with EUS in the lymphadenopathy category, Is this correct?

Response

Thank you for your comment. The significant difference in EUS detection of lymphadenopathy in Table 2 is due to the higher number of cases without EUS in the benign group. There were no cases of lymphadenopathy associated with IPMN detected in this study, so this significant difference does not hold any substantive meaning. We have added this explanation to the Results section.

 P16 Line 236-238

“Additionally, the significant differences in cyst size and lymphadenopathy detected by EUS are due to the higher number of cases without EUS in the benign group. These significant differences do not have substantive meaning.”

7. How was the type of IPMN classified in preoperative examinations? If the classification is based on the main pancreatic duct diameter, then both the IPMN type and the main pancreatic duct diameter are included as the same indicator in the score. I recommend to include only one of them in the score.

Response

Thank you for the important feedback. Based on Reviewer 1's suggestions, we conducted multivariate analysis to enhance the accuracy of the scores. However, as you noted, the type of IPMN exhibited multicollinearity with the main duct size. We then categorized IPMN based on the presence or absence of cysts, combining branch-duct and mixed types into a single category while treating the main duct type separately. As a result, neither univariate nor multivariate analysis revealed a significant difference. Additionally, including IPMN type in the scoring system did not improve the AUC, so we have decided to exclude it from the scoring system.

Reviewer #3: 

The issue reports a very interesting new approach for IPMN scoring model. Even if it was based on a retrospective analysis, either the method or the results seem to be enchouraging to have an improvement of better relation between preoperative and pathological finndings. In particular IMAP model score 6 showed strong specificity and acceptable sensitivity. I think this study should be considered a very concrete step forward in order to assess a useful and practice model in management of IPMN likelihood risk staging.

Response

Thank you for your kind words. They are very encouraging and much appreciated.

Reviewer #4: 

The authors describe a single institutional retrospective study analyzing IPMN cases focusing on the presence of malignancy in those tumors. Also, the authors developed the scoring system to predict the presence of malignancy in IPMN cases (IMAP scoring system). Their scoring system includes Age, IPMN type, DM status, CA19-9, enhancing mural nodule, thickened cyst wall, MPD size, and jaundice. The authors evaluated the significance of IMAP scoring system by calculating AUC, and other parameters (e.g., sensitivity, sp

---

## [Decision Letter · Decision Letter 1]

4 Oct 2024

Development of an intraductal papillary mucinous neoplasm malignancy prediction scoring system

PONE-D-24-16942R1

Dear Dr. Kobayashi,

We’re pleased to inform you that your manuscript has been judged scientifically suitable for publication and will be formally accepted for publication once it meets all outstanding technical requirements.

Kind regards,

Fabrizio D'Acapito, Ph.D,M.D.

Academic Editor

PLOS ONE

Additional Editor Comments (optional):

Three of the four reviewers consulted found the study suitable for publication.

I have carefully read the new draft of the manuscript and, although I understand the concerns of reviewer n.4, I believe that the authors have done an adequate job of improving the text.

I hope that their score can be properly validated in the near future.

Congratulations

Reviewers' comments:

Reviewer's Responses to Questions

**Comments to the Author**

1. If the authors have adequately addressed your comments raised in a previous round of review and you feel that this manuscript is now acceptable for publication, you may indicate that here to bypass the “Comments to the Author” section, enter your conflict of interest statement in the “Confidential to Editor” section, and submit your "Accept" recommendation.

Reviewer #1: All comments have been addressed

Reviewer #2: All comments have been addressed

Reviewer #4: (No Response)

2. Is the manuscript technically sound, and do the data support the conclusions?

Reviewer #1: Yes

Reviewer #2: Yes

Reviewer #4: No

3. Has the statistical analysis been performed appropriately and rigorously? 

Reviewer #1: Yes

Reviewer #2: Yes

Reviewer #4: No

4. Have the authors made all data underlying the findings in their manuscript fully available?

Reviewer #1: Yes

Reviewer #2: Yes

Reviewer #4: Yes

5. Is the manuscript presented in an intelligible fashion and written in standard English?

Reviewer #1: Yes

Reviewer #2: Yes

Reviewer #4: Yes

6. Review Comments to the Author

Reviewer #1: The authors have answered the comments. The manuscript has been improved. I have no additional comments.

Reviewer #2: I have reviewed the revised paper and it has been properly corrected.Additional point has not been found.

Reviewer #4: The authors did not appropriately respond to my comments.

In the authors’ response, the authors mentioned that “Our study focused on differentiating between benign and malignant cases pathologically confirmed to be IPMN postoperatively.”.

However, at the same time, the clinical application of this system is described as “it is crucial to use this scoring system preoperatively”. We must question the appropriateness of adapting a system optimized for cases with postoperative histopathologic diagnosis of IPMN to those without a histopathologic diagnosis of IPMN. Indeed, the authors also mentioned that “Cases that might have been evaluated as malignant if resected and pathologically assessed cannot be conclusively classified as malignant without resection.”. In this regard, a significant number of malignant cases may have been excluded from the analysis. Therefore, it is quite difficult appropriately interpret the results of this study and the rationale for employing this system in preoperative evaluation cannot be firmly established. At a minimum, authors should include all cases with WF or above, both resected and non-resected, in their analysis.

Regarding the operative eligibility, the authors indicated in the response that “we adhere to the International Consensus Guideline for the management of IPMN. We use CT/MRI or EUS to screen for WF/HRS, and surgery is recommended if HRS is confirmed.”. The frameworks of WF and HRS were introduced in the 2012 international consensus guidelines and have undergone subsequent revisions. Considering the study period extended from 2008 to 2022, it is plausible to assume that variations in the definitions of WF and HRS resulted in inconsistencies in surgical indications during this time. Based on these findings, I believe that all IPMN cases, irrespective of resection status, should be incorporated into the analysis.

7. PLOS authors have the option to publish the peer review history of their article (what does this mean?). If published, this will include your full peer review and any attached files.

Reviewer #1: No

Reviewer #2: No

Reviewer #4: No

---

## [Editor Report · Acceptance letter]

7 Oct 2024

PONE-D-24-16942R1 

PLOS ONE

Dear Dr. Kobayashi, 

I'm pleased to inform you that your manuscript has been deemed suitable for publication in PLOS ONE. Congratulations! Your manuscript is now being handed over to our production team.

Kind regards, 

on behalf of

Dr. Fabrizio D'Acapito 

Academic Editor

PLOS ONE